# Enhancing Interpretability in Deep Reinforcement Learning through Semantic Clustering

## Abstract

In this paper, we explore semantic clustering properties of deep reinforcement learning (DRL) to improve its interpretability and deepen our understanding of the internal semantic organization. In this context, semantic clustering refers to the ability of neural networks to cluster inputs based on their semantic similarity in the internal space. We propose a DRL architecture that incorporates a novel semantic clustering module, which includes both feature dimensionality reduction and online clustering. This module integrates seamlessly into the DRL training pipeline, addressing the instability of t-SNE and eliminating the need for extensive manual annotation in the previous semantic analysis methods. Through experiments, we validate the effectiveness of the proposed module and demonstrate its ability to reveal semantic clustering properties within DRL. Furthermore, we introduce new analytical methods that leverage these properties to provide insights into the hierarchical structure of policies and the semantic organization within the feature space. These methods also help identify potential risks within the model, offering a deeper understanding of its limitations and guiding future improvements.

## 1 Introduction

Deep reinforcement learning (DRL) has been widely applied in various domains such as robotics, autonomous systems, game playing, and healthcare, due to its ability to solve complex decision-making tasks (Ernst & Louette, 2024; Li, 2023; Shakya et al., 2023). However, the black-box nature of DRL models makes the decision-making process difficult to understand, potentially leading to unforeseen risks. In this study, we explore semantic clustering properties to improve the interpretability of DRL models. *Semantic clustering* refers to the process of grouping information based on underlying semantic similarities in the feature space of neural networks. Studying semantic clustering in DRL helps uncover the model's internal knowledge structure and reveals semantic relationships between states, enhancing both the interpretability and transparency of DRL models.

Although semantic clustering has been thoroughly investigated in natural language processing (NLP) (Rong, 2014; Pennington et al., 2014) and computer vision (CV) (Long et al., 2023; Saha et al., 2023; Park et al., 2021; Zhou & Wei, 2020), it remains underexplored in DRL, largely due to the complexity introduced by temporal dynamics and the absence of direct supervised signals. The sequential nature of decisions in DRL further complicates the task of capturing evolving semantics. Early work introduced *external* constraints—e.g., bisimulation (Kemertas & Aumentado-Armstrong, 2021; Zhang et al., 2021) and contrastive learning (Eysenbach et al., 2022; Agarwal et al., 2021; Laskin et al., 2020)—to shape feature spaces conducive to semantic clustering. In contrast, we focus on investigating whether DRL can *intrinsic* exhibit semantic clustering capabilities.

Mnih et al. (2015) and Zahavy et al. (2016) analyzed the semantic distribution of the DRL feature space for Atari games using t-distributed Stochastic Neighbor Embedding (t-SNE) (van der Maaten & Hinton, 2008). Nevertheless, these studies are limited in multiple ways: (i) they are limited to a

small set of Atari games with fixed scenes, making it difficult to distinguish whether clustering arises from pixel similarity or semantic understanding, (ii) Zahavy et al. (2016) manually define features for specific games, limiting the adaptability of their method to other environments, and (iii) both studies rely on t-SNE visualization for semantic analysis, which tends to produce unstable results and lacks an automated clustering mechanism. Consequently, these approaches require significant manual effort for feature space annotation and analysis, hindering comprehensive semantic analysis and integration into downstream tasks.

Specifically, we make the following key contributions:

- We perform a comprehensive exploration of semantic clustering properties in DRL, advancing the understanding of the black-box decision-making processes. In contrast to prior work that uses a small set of Atari games with fixed scenes, we use Procgen [1] (Cobbe et al., 2020), which offers rich semantic diversity and dynamic, procedurally generated environments, making it well-suited for investigating semantic clustering in DRL.

- We introduce a novel architecture that addresses the limitations in previous semantic analysis methods based on t-SNE, providing a more stable and effective approach for studying semantic properties in DRL.

- We present new methods for model and policy analysis, which help to understand the internal semantic distribution, the hierarchical structure of policies, and identify potential risks in the DRL model.

## 2 RELATED WORK

**Semantic Clustering in NLP and CV**  Prior work in NLP has shown that the spatial arrangement of word embeddings reflects semantic similarities, with semantically-related terms forming clusters in the embedding space (Pennington et al., 2014; Rong, 2014). Similarly, in computer vision, images with similar content are positioned closely in the learned feature space (Long et al., 2023; Saha et al., 2023; Park et al., 2021; Zhou & Wei, 2020).

**Semantic Clustering in DRL**  Mnih et al. (2015) and Zahavy et al. (2016) have previously explored visualizing the DRL feature space using t-SNE. In these studies, t-SNE visualizations show that features of states with close pixel distances tend to cluster together. However, due to the fixed nature of the scenarios they used (Atari games), semantic clustering could not be conclusively verified. This limitation motivates our use of Procgen to validate our approach.

**Interpretability of DRL**  DRL interpretability research often focuses on video games due to their controlled environments and clear rules, which make analyzing decision-making processes easier. PW-Net (Kenny et al., 2023) enhances interpretability by using human-friendly prototypes to explain the model's decision-making. DIGR (Xing et al., 2023) improves interpretability by generating saliency maps that highlight the most relevant features influencing the agent's decisions. Concept policy models integrate expert knowledge into multi-agent RL, enabling real-time intervention and clearer interpretation of agent behavior (Zabounidis et al., 2023). MENS-DT-RL (Costa et al., 2024) applies decision trees to provide a transparent, rule-based explanation of the learning process. Furthermore, attention mechanisms and symbolic reasoning frameworks have also been applied to enhance interpretability (Shi et al., 2022; Mott et al., 2019; Lyu et al., 2019). Our work explores how DRL models internally organize information, offering new insights into the structure of learned representations.

## 3 METHOD

Our proposed architecture with a novel semantic clustering module is presented in Figure 1.

---

[1]Detailed environment instructions are available here, in Appendix A of Cobbe et al. (2020), and in the Procgen repository.

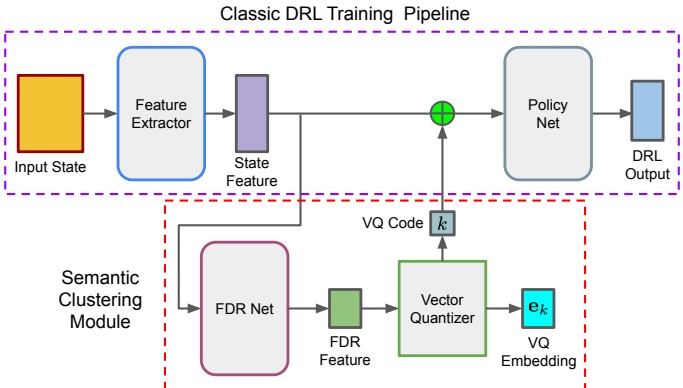

Figure 1: Overview of our architecture. The upper segment represents the classic DRL training pipeline, while the lower segment introduces the semantic clustering module. The Feature Dimensionality Reduction (FDR) net reduces the dimensionality of state features, resulting in FDR features, which the vector quantizer then processes to generate discrete VQ codes (denoted $k$), which represent states associated with clusters, along with the closest VQ embeddings. Subsequently, $k$ is integrated into the state feature by element-wise addition after being expanded to match the state feature dimensions, enabling conditional policy training that better supports the integration of downstream tasks.

**Background** Vector-quantized variational autoencoders (VQ-VAE) (van den Oord et al., 2017) are a family of successful generative models that combine the variational autoencoder (VAE) framework with discrete latent representations through a novel parameterization of the posterior distribution. The VQ-VAE workflow begins with an encoder network $\hat{E}$ that maps an input $\mathbf{x}$ to a latent representation $\hat{E}(\mathbf{x})$. This representation is then quantized by mapping it to the nearest embedding in a codebook $\{\mathbf{e}_k | k \in \{1, 2, \ldots, K\}\}$. The quantized representation is then passed into a decoder network $\hat{D}$ to reconstruct the input $\mathbf{x}$. The loss function for VQ-VAE is defined as:

$$\mathcal{L}_{\text{VQ-VAE}} = \left\| \mathbf{x} - \hat{D}(\mathbf{e}_k) \right\|_2^2 + \left\| sg\left( \hat{E}\left(\mathbf{x}\right) \right) - \mathbf{e}_k \right\|_2^2 + \beta \left\| sg\left(\mathbf{e}_k\right) - \hat{E}\left(\mathbf{x}\right) \right\|_2^2, \tag{1}$$

where $sg$ is a stop-gradient operator and $\beta$ weights the distance reduction between the encoded output $\hat{E}(\mathbf{x})$ and its closest embedding $\mathbf{e}_k$.

In this paper, we modify VQ-VAE to achieve the following objectives: (i) Assign features to the nearest VQ embedding for clustering. (ii) Seamlessly integrate with DRL training, enabling simultaneous clustering and policy learning. (iii) Enhance clustering and interpretability through joint training with additional losses. Further details are provided in § 3.2.

## 3.1 SEMANTIC CLUSTERING MODULE

To overcome the limitations of previous t-SNE-based semantic analyses (see § 1), we propose a novel semantic clustering module, which includes dimensionality reduction and online clustering.

**Dimensionality Reduction** Given the complexity of states in most DRL applications, their features are often high-dimensional. For example, DQN (Mnih et al., 2015) uses 512-dimensional features when trained on Atari games. Clustering high-dimensional features is challenging due to the "curse of dimensionality," which diminishes the effectiveness of distance metrics as dimensions increase, making it difficult to assess data point similarity accurately (Beyer et al., 1999; Aggarwal et al., 2001). To mitigate these issues, we reduce feature dimensionality before clustering, resulting in more robust and interpretable clustering outcomes. This not only simplifies the clustering process but also enables human-interpretable visualizations, typically in 2D.

The instability of t-SNE arises from its non-convex objective function, making it highly sensitive to initialization and leading to varied visualization outcomes (van der Maaten & Hinton, 2008; Wattenberg et al., 2016). To overcome these challenges, we propose the *Feature Dimensionality Reduction* (FDR) network. This network remaps high-dimensional features to 2D using policy

training data for online training, ensuring stable and efficient mappings. The FDR network's loss function is designed to preserve the consistency of *distance relationships* between high-dimensional and 2D feature spaces, measured by pairwise similarities as described in § 3.2.

**Online Clustering**  To reduce the labor involved in extensive manual annotation from previous work and to facilitate the integration of downstream tasks, we propose using online clustering to segment the feature space and assist in semantic analysis. We modify the VQ-VAE framework to achieve online clustering, as detailed in § 3.2.

## 3.2 LOSS FUNCTION DESIGN

The loss function for our proposed framework is given by

$$\mathcal{L}_{\text{total}} = \mathcal{L}_{\text{DRL}} + f_{\text{control}} \left( w_{\text{FDR}} \mathcal{L}_{\text{FDR}} + w_{\text{VQ-VAE}} \mathcal{L}'_{\text{VQ-VAE}} \right). \tag{2}$$

The DRL loss function $\mathcal{L}_{\text{DRL}}$ comes from the original DRL model. $w_{\text{FDR}}$ and $w_{\text{VQ-VAE}}$ are the weights of the FDR loss ($\mathcal{L}_{\text{FDR}}$) and the modified VQ-VAE loss ($\mathcal{L}'_{\text{VQ-VAE}}$), respectively. $f_{\text{control}}$ represents the control factor. We explain each of these components below.

**FDR Loss**  The FDR loss function is based on state features from the training batch and the FDR features generated by the FDR network. Here, we choose the Student's t-distribution to measure the pairwise similarities for two key reasons: (i) it more effectively captures the nonlinear structure of data and preserves local neighborhood relationships compared to methods like PCA (Abdi & Williams, 2010; Lee et al., 2004) and UMAP (McInnes et al., 2018), and (ii) it efficiently measures pairwise relative positional relationships between features within a batch, without needing to observe the entire feature set, making it well-suited for online clustering.

The pairwise similarities of state features $p_{ij}$ is given by

$$p_{ij} = \frac{d(i,j)}{\sum_{k \neq l} d(k,l)}, \text{ where } d(m,n) = \left( 1 + \frac{\|f(\mathbf{s}_m) - f(\mathbf{s}_n)\|^2}{\alpha} \right)^{-\frac{\alpha+1}{2}} \tag{3}$$

where $f$ is the feature extractor, $\mathbf{s}_i$ is the $i^{\text{th}}$ state in a batch, and $\alpha$ is the degree of freedom. The pairwise similarities for FDR features, $q_{ij}$ are computed using the same expression as equation 3, but with $f$ replaced by $g \circ f$, where $g$ is the FDR net. In contrast to other deep clustering studies, e.g., Xie et al. (2016) and Li et al. (2018), the same degree of freedom $\alpha$ is selected for both high- and low-dimensional similarities, ensuring that the original distance relationship between features is maintained in the low-dimensional space.

The FDR loss is given by

$$\mathcal{L}_{\text{FDR}} = -\sum_i \sum_j p_{ij} \log(q_{ij}). \tag{4}$$

**Modified VQ-VAE Loss**  The first term in $\mathcal{L}_{\text{VQ-VAE}}$ from equation 1 supports input reconstruction, which our model does not need. The third term causes FDR features to align with their nearest VQ embeddings. The third term ensures that FDR features align with their nearest VQ embeddings, while the second term moves VQ embeddings closer to neighboring FDR features. We leverage the property of the second term to group FDR features effectively. These embeddings function similarly to centroids in online $k$-means (MacQueen, 1967). Therefore, we eliminate the first and third terms and modify the second term to obtain the modified VQ-VAE loss:

$$\mathcal{L}'_{\text{VQ-VAE}} = \|sg\left[g(f(\mathbf{s}))\right] - \mathbf{e}_k\|_2^2, \tag{5}$$

where $sg$ is the stop-gradient operation, and $\mathbf{e}_k$ is the closest embedding in the codebook to the FDR feature $g(f(\mathbf{s}))$.

**Control Factor**  Since effective semantic clustering relies on a clear and distinguishable semantic distribution that is often difficult to achieve in the early stages of training, we propose an adaptive control factor strategy updated according to training performance (see Appendix A).

**Improved Clustering**   Our loss design not only achieves dimensionality reduction and clustering but also enhances the clustering properties of the feature space, making states within each cluster more compact and the cluster boundaries more separable. This improvement is crucial for clearly distinguishing the semantics of states at the cluster boundaries, further enhancing the model's interpretability. We demonstrate the improved clustering in § 4 and provide more experimental evidence of this enhanced clustering property and the intrinsic nature of semantic clustering in DRL in Appendix C.

This enhanced clustering arises from the fact that although the stop-gradient operation ($sg$) in $\mathcal{L}'_{\text{VQ-VAE}}$ prevents FDR features from moving towards the closest VQ embeddings, when the FDR features become denser under the influence of $\mathcal{L}_{\text{FDR}}$, it indirectly reduces $\mathcal{L}'_{\text{VQ-VAE}}$ and, consequently, $\mathcal{L}_{\text{total}}$. In other words, training with both $\mathcal{L}_{\text{FDR}}$ and $\mathcal{L}'_{\text{VQ-VAE}}$ decreases the sum of distances between FDR features and their corresponding VQ embeddings within the same cluster. Additionally, since $\mathcal{L}_{\text{FDR}}$ aims to bring $p$ and $q$ closer, the improved cluster density in the FDR feature space is also transferred to the state feature space.

---

**Algorithm 1** PPO with Semantic Clustering Module (SCM)

---

1: **Initialize:**
- PPO network parameters $\theta$
- FDR network parameters $\phi$
- Vector quantizer codebook $\{\mathbf{e}_k \mid k = 1, 2, \ldots, K\}$
- SCM hyperparameters: $w_{\text{FDR}}$, $w_{\text{VQ-VAE}}$, $f_{\text{control}}$
- PPO hyperparameters: value loss weight $w_{\text{value}}$, entropy loss weight $w_{\text{entropy}}$, and others such as discount factor $\gamma$, clipping range $\epsilon$, and batch size $M$

2: **for each** training iteration $i = 1, 2, \ldots$ **do**
3:     Collect $N$ trajectories $\mathcal{D}_i = \{\tau_1, \tau_2, \ldots, \tau_N\}$ by running the policy $\pi_\theta$ in the environment.
4:     **for each** epoch $j = 1, 2, \ldots$ **do**
5:        **for each** minibatch $M \subseteq \mathcal{D}_i$ **do**
6:           **for each** state $\mathbf{s}_m \in M$ **do**
7:              $\mathbf{f}_m \leftarrow f_\theta(\mathbf{s}_m)$.         ▷ *Extract the state feature using the feature extractor*
8:              $\mathbf{f}_m^{\text{FDR}} \leftarrow g_\phi(\mathbf{f}_m)$.         ▷ *Extract the FDR feature using the FDR net*

9:              ▷ *Assign the FDR feature to the nearest embedding in the vector quantizer*     ◁
10:             $k_m \leftarrow \arg\min_k \|\mathbf{f}_m^{\text{FDR}} - \mathbf{e}_k\|, \quad \mathbf{e}_k \in \{\mathbf{e}_1, \mathbf{e}_2, \ldots, \mathbf{e}_K\}$.

11:              ▷ *Expand $k_m$ to match state feature dimensions and add it to the state feature* ◁
12:             $\mathbf{k}_m^{\text{expand}} \leftarrow \text{expand}(k_m, \dim(\mathbf{f}_m))$.
13:             $\mathbf{f}_m^{\text{fused}} \leftarrow \mathbf{f}_m + \mathbf{k}_m^{\text{expand}}$.

14:              ▷ *Compute the policy and value outputs using the policy layers $\hat{\pi}_\theta$ and the value layers $\hat{V}_\theta$ in the PPO network*     ◁
$$\pi(a|\mathbf{s}_m) \leftarrow \hat{\pi}_\theta(\mathbf{f}_m^{\text{fused}}), \quad V(\mathbf{s}_m) \leftarrow \hat{V}_\theta(\mathbf{f}_m^{\text{fused}}).$$

15:           ▷ *Compute the policy loss $\mathcal{L}_{\text{policy}}$, value loss $\mathcal{L}_{\text{value}}$, and entropy loss $\mathcal{L}_{\text{entropy}}$ based on the PPO algorithm, then combine the losses into the PPO loss*     ◁
16:           $\mathcal{L}_{\text{PPO}} \leftarrow \mathcal{L}_{\text{policy}} + w_{\text{value}} \cdot \mathcal{L}_{\text{value}} + w_{\text{entropy}} \cdot \mathcal{L}_{\text{entropy}}$.
17:           $\mathcal{L}_{\text{SCM}} \leftarrow w_{\text{FDR}} \cdot \mathcal{L}_{\text{FDR}} + w_{\text{VQ-VAE}} \cdot \mathcal{L}'_{\text{VQ-VAE}}$.         ▷ *Compute the SCM loss*
18:           $\mathcal{L}_{\text{total}} \leftarrow \mathcal{L}_{\text{PPO}} + f_{\text{control}} \cdot \mathcal{L}_{\text{SCM}}$.         ▷ *Compute the total loss*
19:           Update $\theta$, $\phi$, and $\{\mathbf{e}_k\}_{k=1}^{K}$ by minimizing $\mathcal{L}_{\text{total}}$.         ▷ *Optimize model*

---

**Advantages of Online Training**   Online training offers several key advantages: (i) It enhances clustering by incorporating the training of $\mathcal{L}_{\text{total}}$. (ii) Training the VQ code $k$ with a latent-conditioned policy $\pi(a|\mathbf{s}, k)$ (where $a$ is the action) supports the extension to downstream tasks, such as macro action selection in hierarchical learning. (iii) Online training improves memory efficiency by eliminating the need to store a large number of states during model training.

**Training Process** The training process of the proposed framework builds upon the structure of the original DRL algorithm while incorporating the semantic clustering module (SCM) by using equation 2 for total loss calculation. We take PPO (Schulman et al., 2017) as an example, and the detailed training procedure is outlined in Algorithm 1.

# 4 SIMULATIONS

In this section, we aim to: (i) validate the clustering effectiveness of our proposed method, (ii) assess the semantic clustering properties of DRL in Procgen, and (iii) introduce new methods for policy and model analysis.

## 4.1 CLUSTERING EFFECTIVENESS EVALUATION

We demonstrate the clustering effectiveness of our proposed approach using the CoinRun game from Procgen as an example. Similar results can be easily extended to other games using the code and checkpoints provided in the supplementary material. We use a trained model to collect states, where the agent selects actions randomly with a probability of 0.2 and follows the trained policy with a probability of 0.8 to ensure diverse state coverage. States are sampled with a probability of 0.8, and 64 parallel environments collect states over 500 steps, resulting in approximately 25,000 states for visualization.

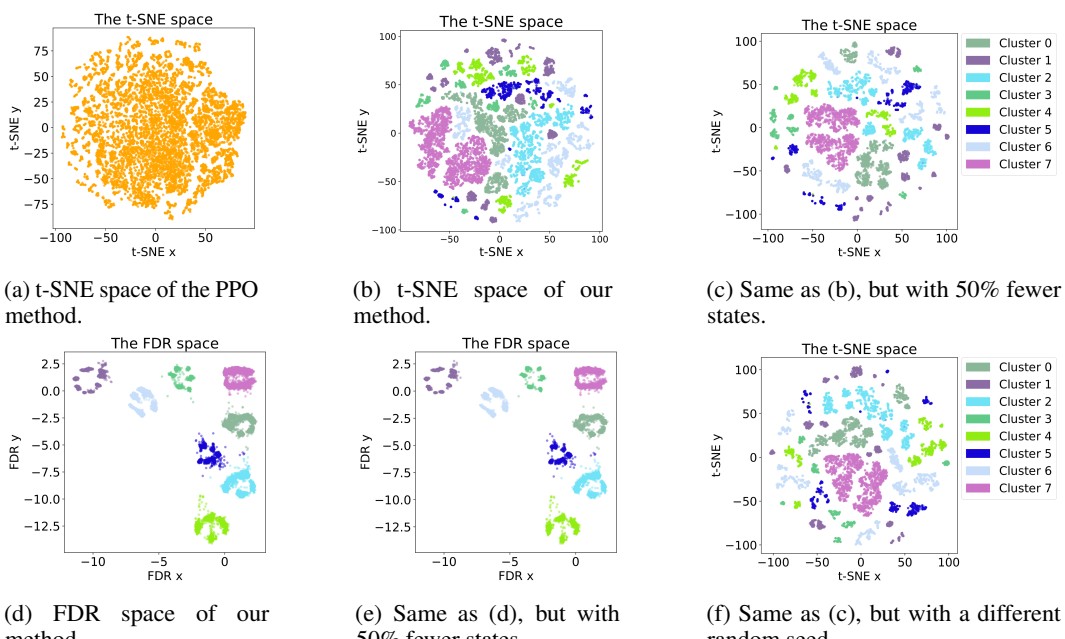

(a) t-SNE space of the PPO method.

(b) t-SNE space of our method.

(c) Same as (b), but with 50% fewer states.

(d) FDR space of our method.

(e) Same as (d), but with 50% fewer states.

(f) Same as (c), but with a different random seed.

Figure 2: Visualization of features in t-SNE and FDR spaces, using PPO and our method. To facilitate comparing spatial relationships and changes in feature distributions, feature colors in the t-SNE visualizations of our method correspond to the cluster colors in the FDR space. Note that PPO-generated features are uniformly colored orange as they lack cluster information.

**Cluster Separation and Improved Clustering** The t-SNE visualization of PPO (Figure 2a), spreads features across the space without forming clear clusters, limiting its utility for clustering analysis and requiring detailed manual examination of certain areas, as in previous studies. In contrast, the t-SNE visualization of our method (Figure 2b), reveals numerous distinct, small clusters. States within each of these clusters originate from the same semantic group identified by our method. This dispersion into multiple smaller clusters is due to t-SNE's focus on local structures and its tendency to avoid crowding, causing complete semantic clusters to scatter. The visualization in the FDR space (Figure 2d), displays clear and separate complete clusters, which are identified by VQ codes.

**Sensitivity to Number of States**   Our method's stability is showcased in Figures 2c and 2e, where the number of processed states is reduced by 50%. Unlike the drastic changes in feature distribution seen in the t-SNE space (Figure 2c), the FDR space (Figure 2e) exhibits a stable mapping, merely reducing the quantity of features without altering their spatial distribution.

**Sensitivity to Random Seed**   While the t-SNE representation is sensitive to randomness, as demonstrated by the significant difference between Figures 2c and 2f, the FDR space's mapping remains unchanged even when the random seed is altered, maintaining the distribution in Figure 2e. t-SNE's randomness primarily stems from its random initialization and non-convex optimization process, leading to different visualizations with different random seeds. In contrast, our model produces a stable feature mapping after training, which does not vary with random seeds.

These stark contrasts underscore the FDR space's mapping robustness compared to t-SNE's instability, addressing the limitations of previous t-SNE-based analysis methods and laying a solid foundation for stable semantic clustering analysis and our proposed policy and model analysis.

## 4.2   SEMANTIC CLUSTERING IN DRL

In this section, we illustrate semantic clustering analysis using the Ninja game, in which the agent goes from left to right, jumping over various ledges and scores points by touching the mushroom on the far right. In Appendix D, we analyze additional games, reaching similar conclusions.

**Mean Image Analysis**   We performed a qualitative analysis of the mean images of states within each cluster. Figure 3 presents state examples from the FDR space of Ninja along with the mean images of each semantic cluster, and Table 1 contains natural language descriptions of the clusters as well as notable features of the mean images corresponding to each cluster. Corresponding videos can be found in the supplementary material.

Unlike *static* semantic clustering in some CV and NLP tasks, where clustering is based on a single image or word, DRL's semantic clustering is *dynamic* in nature—state sequences with similar semantics are grouped into the same semantic cluster. Notably, this semantic clustering goes beyond pixel distances and operates on a *semantic understanding* level of the environment, as illustrated in figures 3 and 4. This generalized semantic clustering emerges from the DRL model's inherent ability to learn and summarize from changing scene dynamics, independent of external constraints like bisimulation or contrastive learning, and without the need for supervised signals. The neural network's internal organization of policy-relevant knowledge indicates clustering-based spatial organization based on semantic similarity. Furthermore, we find that video sequences within clusters can be summarized using natural language, akin to the 'skills' humans abstract during learning processes.

**Human Evaluation**   In addition to qualitatively analyzing the mean images, we hired 15 human evaluators to validate the semantic clustering properties. Specifically, video sequences from each episode are segmented into multiple clips based on the cluster each frame belongs to, and these clips are grouped by cluster for evaluators to review. Each evaluator watched these grouped clips and responded to three interpretability-related statements for two out of a set of three games (Jumper, Fruitbot, and Ninja). The response for each question was chosen from a five-point Likert scale with the following options: *Strongly Disagree (1)*, *Disagree (2)*, *Neutral (3)*, *Agree (4)*, and *Strongly Agree (5)*. Further details on the human evaluation procedure are provided in Appendix G. The statements and the results of the human evaluation are provided in Table 2. The mean scores for all statement-environment combinatiosn are greater than 4, with the exception of statements 1 and 3 for FruitBot, for which the lower bounds on the mean set by the standard error of the mean (SEM) are 3.99 and 3.92 respectively. One explanation for the slightly lower score on FruitBot is that in the description of behaviors within clusters in FruitBot, the agent's relative distance to the wall ahead (far/near) and the agent's relative position on the screen (left/center/right) require a higher degree of subjective judgment—in contrast, Jumper has a clear radar for direction and position information, and Ninja has more explicit behavioral reference objects, e.g., ledges and mushrooms. These results suggest that humans generally agree that our model possesses semantic clustering properties and supports interpretability.

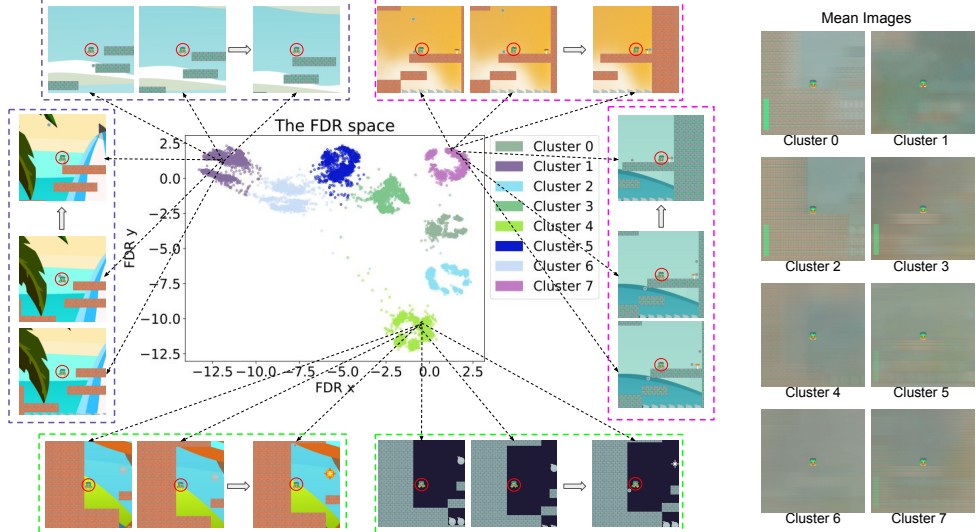

Figure 3: State examples in the Ninja FDR space and the mean images of clusters. Each dashed box contains a set of consecutive states from the same cluster. The dotted arrows indicate the positions of the FDR features corresponding to the states. Descriptions of the state sequences in the clusters are provided in Table 1.

Table 1: Cluster descriptions and mean image outlines for the Ninja game

| Cluster | Description | Mean image outlines |
|---|---|---|
| 0 | The agent starts by walking through the first platform and then performs a high jump to reach a higher ledge. | Essential elements are outlined, e.g., a left-side wall, the current position of the agent on the first platform, and the upcoming higher ledges. |
| 1 | The agent makes small jumps in the middle of the scene. | We can observe the outlines of several ledges below the agent. |
| 2 | Two interpretations are present: 1) the agent starts from the leftmost end of the scene and walks to the starting position of Cluster 0, and 2) when there are no higher ledges to jump to, the agent begins from the scene, walks over the first platform, and prepares to jump to the subsequent ledge. | The scene prominently displays the distinct outline of the left wall and the first platform. The agent's current position is close to both of them. |
| 3 | The agent walks on the ledge and prepares to jump to a higher ledge. | The agent is standing on the outline of the current ledge and the following higher ledges. |
| 4 | After performing a high jump, the agent loses sight of the ledge below. | The agent is performing a high jump. |
| 5 | The agent walks on the ledge and prepares to jump onto a ledge at the same height or lower. | The agent is standing on the outline of the current ledge and the following ledges at the same height or lower. |
| 6 | The agent executes a high jump while keeping the ledge below in sight. | The agent is performing a high jump and the outline of the ledge below is visible. |
| 7 | The agent moves towards the right edge of the scene and touches the mushroom. | The outlines of the wall and platform on the far right are visible. |

Table 2: Human evaluation results

| No. | Statement | Mean Score (SEM) | | |
|---|---|---|---|---|
| | | Jumper | FruitBot | Ninja |
| 1 | *The clips of each cluster consistently display the same skill being performed* | 4.24 (0.15) | 4.10 (0.11) | 4.30 (0.15) |
| 2 | *The clips of each cluster match the given skill description* | 4.36 (0.16) | 4.16 (0.11) | 4.20 (0.17) |
| 3 | *The identified skills aid in understanding the environment and the AI's decision-making process* | 4.50 (0.22) | 4.10 (0.18) | 4.20 (0.20) |

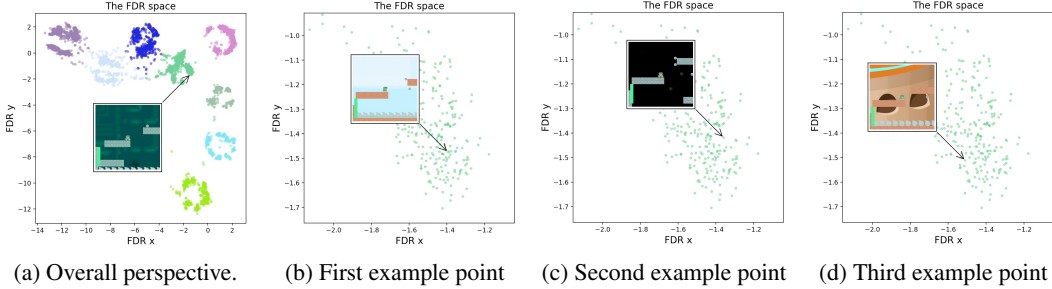

|      Cluster 2      |      Cluster 4      |      Cluster 3      |      Cluster 7      |

Figure 4: Three episodes from the Ninja game. States within colored dashed boxes correspond to the clusters of the same colors in Figure 3. Solid gray arrows indicate omitted states from the same cluster. Ellipses represent other omitted states.

## 4.3 MODEL AND POLICY ANALYSIS

To better explore the knowledge organization within the internal space of DRL models, we developed a visualization tool (see Figure 5 for an example). The tool supports 'statically' analyzing the semantic distribution of models—specifically, (i) when the mouse cursor hovers over a specific feature point, the corresponding state image is displayed, and (ii) the tool includes a zooming functionality to observe the semantic distribution of features in detail within clusters.

| (a) Overall perspective. | (b) First example point | (c) Second example point | (d) Third example point |

Figure 5: Hover examples in the FDR space of Ninja. We observe a sub-cluster in the FDR space as an example from the overall perspective (a) and the zoomed-in perspective (b), (c), and (d). The agent is standing on the edge of a ledge. Although the scenarios of (b), (c), and (d) are different, the proposed method effectively clusters semantically consistent features together in the FDR space.

In addition, we propose a more 'dynamic' analysis method—the VQ code enables us to determine the cluster to which the current state belongs, which allows for the semantic segmentation of episodes, as exemplified in Figure 4. Our model excels at breaking down complex policies, thereby shedding light on their inherent hierarchical structures. Moreover, this segmentation is based on semantics, making it understandable to humans and likely to improve interpretability in downstream hierarchical learning tasks. Consequently, this method introduces a 'dynamic' strategy for dissecting policy structures.

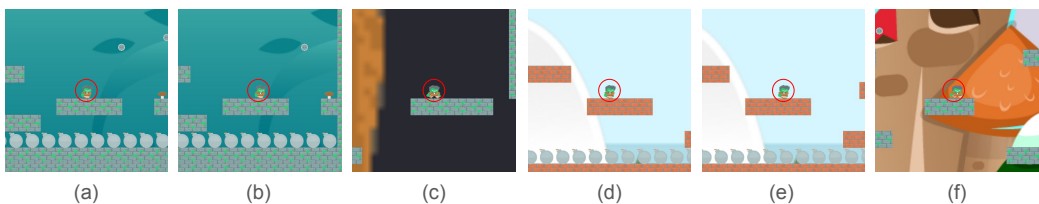

Figure 6: Policy analysis examples in Ninja.

We also present several examples of policy analysis in Figure 6, based on the clustering result from our approach. In 6(a) and 6(b), two consecutive states are shown, with 6(a) and 6(b) belonging to clusters 5 and 7 respectively. We see that determining whether the agent will exhibit the behavior of approaching the mushroom does not depend on the appearance of the mushroom, but rather on the presence of the right-side wall. This is validated in another episode shown in 6(c), which belongs to cluster 7 but lacks the appearance of a mushroom. The agent goes back and forth on the ledge for an extended period, consistently remaining in cluster 7, as it mistakenly interprets the key condition for executing behaviors in cluster 7. 6(d) and 6(e) show two states in the same episode. In 6(d), before a higher ledge comes into view, the agent anticipates jumping onto a lower ledge ahead (i.e., the behavior in cluster 5). However, when the higher ledge appears in 6(e), it adjusts its strategy, opting to jump onto the (safer) higher ledge—i.e., executing the behavior in cluster 3. Similarly, 6(f) shows a state that has been assigned to cluster 3 by our model, helping us anticipate the agent's future behavior of jumping onto the higher ledge. These analyses based on our proposed model contribute to a better understanding of policies, revealing underlying structures and identifying issues.

## 5 LIMITATIONS AND FUTURE WORK

Our approach has the following limitations that we plan to address in future work. First, semantic clustering depends on clear semantic distributions in the FDR space, which can become unstable when policies deviate significantly from optimal behavior, leading to ambiguous groupings. To mitigate this, more robust clustering methods may be needed to ensure stability even under suboptimal policies. Second, as our approach is unsupervised, determining the optimal number of clusters is crucial. Too few clusters can result in mixed semantic interpretations within a cluster, reducing clarity, while too many clusters may lead to fragmented and incoherent groupings. We selected eight clusters to balance interpretability and granularity, but future work could explore adaptive clustering techniques that adjust cluster numbers dynamically based on task complexity. Finally, we plan to extend our approach to additional DRL algorithms, benchmarks, and practical applications, which will help evaluate its generalizability and robustness in more complex environments, and multi-agent settings.

## 6 CONCLUSION

In this paper, we investigated the semantic clustering properties of DRL. Using a novel approach that combines dimensionality reduction and online clustering, we analyzed the internal organization of knowledge within the feature space. Our method provides a stable mapping of feature positions and enhances semantic clustering, revealing meaningful structures in continuous sequences of video game states. We demonstrate that semantic clustering in DRL arises dynamically as the agent interacts with its environment. As the agent explores diverse states during reinforcement learning, it naturally clusters semantically related states based on spatial and temporal relationships. This dynamic clustering exploits regularities in the environment, offering a unique approach compared to the static clustering observations in NLP and CV.

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

# Appendix

## Table of Contents

## A  ARCHITECTURE, HYPERPARAMETERS, AND COMPUTATIONAL COSTS

The training of the proposed method is consistent with the Impala architecture (Espeholt et al., 2018), the PPO algorithm (Schulman et al., 2017), and the hyperparameters used in the Procgen paper (Cobbe et al., 2020). The FDR net is composed of two fully connected layers with 128 and 2 neurons, respectively. The codebook in the vector quantizer has eight embeddings, and the degree of freedom in the FDR loss is 20. The random seeds employed in Figure 2 are 2021 and 2031, while the seeds used in Figure B.1 are 2021, 2022, and 2023. We train all models on one NVIDIA Tesla V100S 32GB GPU. The operating system version is CentOS Linux release 7.9.2009.

In Equation 2 of the main paper, $w_{\text{FDR}}$ and $w_{\text{VQ-VAE}}$ are 500 and 1, respectively. $f_{\text{control}}$ is updated every 50 iterations according to the following expression:

$$f_{\text{control}} = \min \left( \frac{s_{\text{mean}}}{0.8 \cdot s_{\text{highest}}}, 1 \right), \tag{A.1}$$

where $s_{\text{mean}}$ is the mean score of the last 100 episodes in training, and $s_{\text{highest}}$ is the highest score of the environment.

All hyperparameters introduced in our method, except for the number of embeddings, were chosen through performance tuning to optimize the model's overall performance. The number of embeddings in the vector quantizer was determined by ensuring that each cluster maintained a singular semantic interpretation.

## B  PERFORMANCE

Considering the cost of time and computational resources, we opt for training our model on the full distribution of levels in the 'easy' mode. In Figure B.1, a comparison of performance curves between the proposed method and the baseline is presented, where 'SPPO' denotes 'semantic' PPO, i.e., PPO integrated with our proposed semantic clustering module. Consistent with the Procgen paper (Cobbe et al., 2020), given the diversity of episodes during training, a single curve represents both training and testing performance. Across these environments we observe that the proposed method closely aligns with the baseline performance, indicating minimal impact on performance from the introduced module.

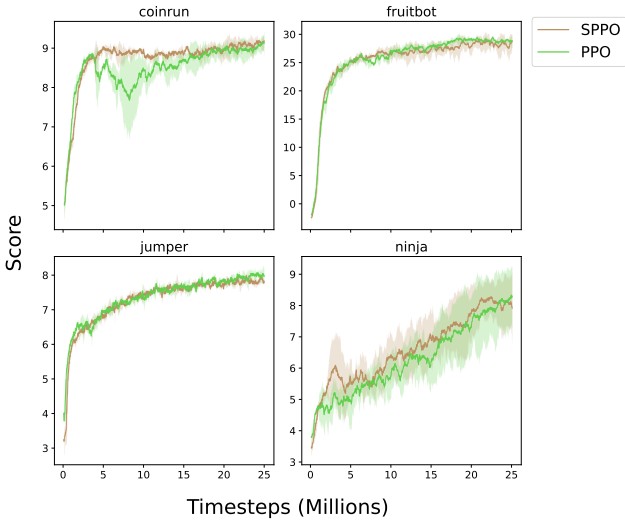

Figure B.1: Performance curves on 'easy' difficulty environments using three random seeds, trained and evaluated on the full distribution of levels.

## C  SEMANTIC CLUSTERING AS AN INTRINSIC PROPERTY OF DRL

We conducted a stop gradient experiment to further investigate whether semantic clustering is an inherent property of DRL. In this experiment, we applied a stop gradient operation to Equation 4 and removed the connection between the VQ codes and the original state features. This was done to prevent the semantic clustering module from influencing the feature space and to observe whether semantic clustering would still occur. The results, as shown in Figures C.1 and C.2, demonstrated that states within the same semantic cluster continued to exhibit similar semantic interpretations, even without the influence of the semantic clustering module. However, the boundaries between clusters became less clear, making it more difficult to distinguish the semantics of states near the edges of clusters.

These observations suggest that semantic clustering is indeed an intrinsic property of DRL, driven by the agent's interaction with its environment during training. The proposed semantic clustering module enhances this natural clustering behavior by increasing the density of clusters, thus improving the separability between them. To fine-tune the influence of the module, we introduced a control factor. At the beginning of training, the control factor is kept low, allowing the DRL training to shape the feature space independently. As the policy becomes more optimized and the semantic distribution of states becomes more organized, the control factor is gradually increased to further enhance the clarity and separability of clusters.

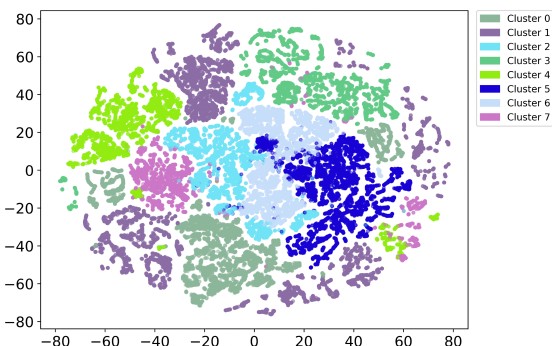

Figure C.1: Visualization of Features in the t-SNE Space. The training eliminates the impact of the proposed semantic clustering module on the original feature space. Feature colors correspond to cluster colors in the FDR space of Figure C.2, facilitating the comparison of spatial relationships and feature distribution changes. Compared to Figure 2b in the main paper, the absence of the semantic clustering module's enhancement makes sub-clusters less distinct.

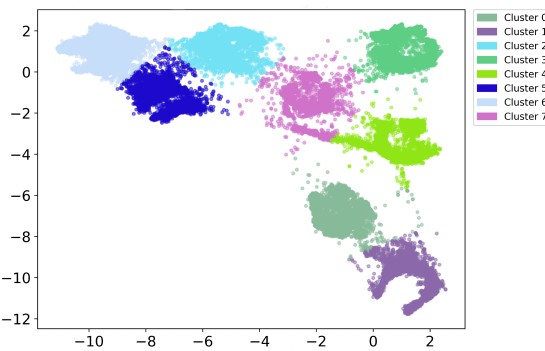

Figure C.2: Visualization of Features in the FDR Space. The training eliminates the impact of the proposed semantic clustering module on the original feature space. Compared to Figure 2d in the main paper, the cluster boundaries in the FDR space are less distinct.

# D  MORE EXAMPLES AND MEAN IMAGES IN THE FDR SPACE

## D.1  COINRUN

To augment the exploration of semantic clustering as discussed in the main paper, this section analyzes two additional games characterized by distinct dynamics. CoinRun's gameplay mechanism is similar to Ninja's, requiring the agent to traverse from the far-left to the far-right, scoring points by interacting with coins at the far-right end of the scene, as illustrated in Figure D.1.

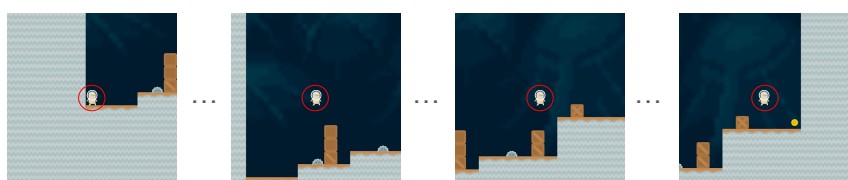

Figure D.1: A episode in CoinRun. Ellipses represent the omitted states.

The observations and insights obtained closely mirror those derived from the analysis of Ninja. Interested readers can leverage the provided code and checkpoint for further exploration of similar findings. Consequently, for brevity, we refrain from extensively elaborating on analogous conclusions.

## D.2 JUMPER

In Jumper, the agent navigates a cave to locate and touch carrots by interpreting a radar displayed in the upper right corner of the screen. The radar's pointer indicates the direction of the carrot, while a bar below the radar shows the distance between the agent and the carrot—shorter bars imply closer proximity, and vice-versa.

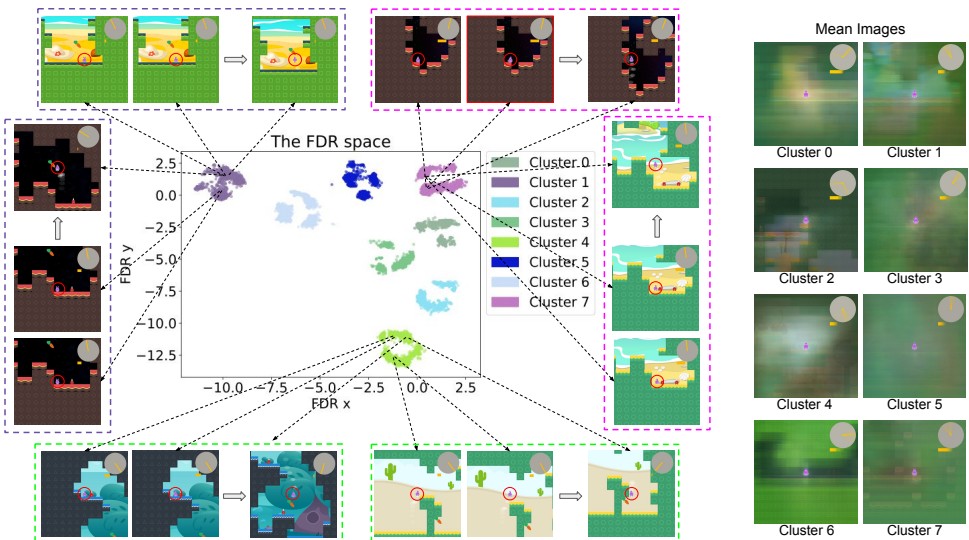

Figure D.2: Examples and mean images from the Jumper FDR space.

The state examples and mean images from the clusters in the FDR space of Jumper are presented in Figure D.2. The background of Jumper is diverse, and the agent is always in the center of the screen (zoom in to see the outlines clearly). In Table 3, we break down descriptions of the sampled images from each cluster and interpretations of the mean image for each cluster in the Jumper game.

Figure D.3 depicts various states from the Jumper game. C.3(a) and C.3(b) belong to the same episode and fall under Cluster 4, while C.3(c) and C.3(d) are from another episode, both categorized under Cluster 1. Notably, neither C.3(a) nor C.3(c) shows the presence of carrots. This observation leads us to suspect that the determination of these clusters is solely reliant on the radar and distance bar rather than the appearance of carrots. To test this hypothesis, we removed the carrot in C.3(e), which originally belonged to Cluster 4, and transformed it into C.3(f). The result demonstrated that C.3(f) still belongs to Cluster 4, confirming our suspicion. However, this phenomenon might pose potential risks in practical applications. For example, in scenarios where sensor data and visual perceptions misalign, AI models might solely rely on sensor data for decision-making—e.g., an autonomous vehicle's sensors indicating an empty road while the occupants inside observe pedestrians crossing, yet the vehicle continues to accelerate.

## D.3 FRUITBOT

FruitBot is a bottom-to-top scrolling game where the agent moves left or right to collect fruits for points while avoiding negative scores upon touching non-fruit objects. The state examples and mean images from the clusters in the FDR space of FruitBot are presented in Figure D.4 and their descriptions in Table 4. FruitBot's mean images lack clarity due to the presence of diverse backgrounds, and the agent is constantly moving to the left and right at the bottom of the screen. However, we can still make out the outline of the wall and agent if we look carefully (zoom in to see the outlines clearly).

Table 3: Cluster descriptions and mean image outlines for the Jumper game

| Cluster | Description | Mean image outlines |
|---|---|---|
| 0 | The agent learns to jump up from the bottom left and move to the left on the top right. | The radar pointing up and to the right, and the outline of the channel above and to the right.. |
| 1 | The agent is touching the carrot on the left or upper left. | The radar is pointing to the upper left and is very close to the target. |
| 2 | The agent learns the skill of movement at the top of the scene. | The radar pointing to the left or down, and the outline of the channels face to the left or down. |
| 3 | The agent is approaching the carrot on the upper right. | The radar pointing to the upper right, and the distance to the target is very close. |
| 4 | The agent is touching the carrot below. | The radar is pointing down, and it is very close to the target. |
| 5 | The agent is approaching the carrot above or left. | The radar is pointing up, and it is very close to the target. |
| 6 | The agent is touching the carrot on the right. | The radar is pointing right, and it is very close to the target. |
| 7 | The agent learns the skill of movement at the right bottom of the scene. | The radar pointing up or to the top left, and it is far from the target. |

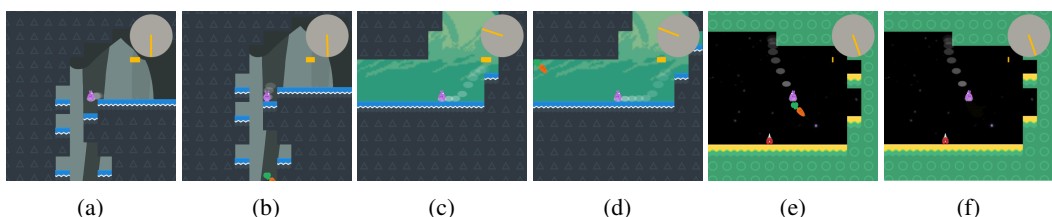

| (a) | (b) | (c) | (d) | (e) | (f) |

Figure D.3: Policy analysis examples in Jumper.

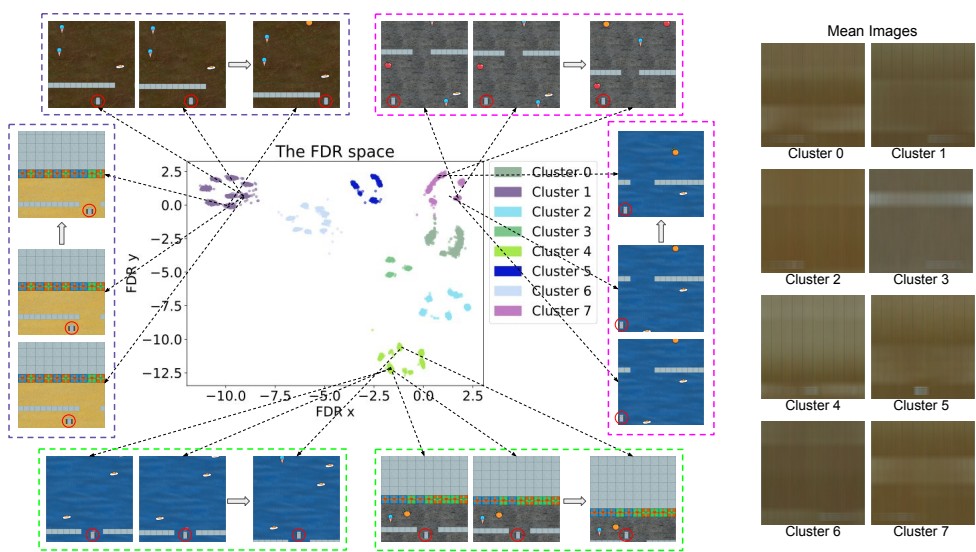

Figure D.4: State examples and mean images from the FruitBot FDR space.

We examined a substantial number of video states and corresponding cluster information, and found that the factors determining clusters in FruitBot are the agent's position on the screen and its relative

Table 4: Cluster descriptions and mean image outlines for the FruitBot game

| Cluster | Description | Mean image outlines |
|---|---|---|
| 0 | The agent is approaching the wall in the left area. | We can see the agent moving toward the gap on the wall that is approaching on the lower left. |
| 1 | The agent approaches the wall from the right area. | The agent is moving toward the gap on the wall that is approaching on the lower right. |
| 2 | The agent executes its policy far from the wall from the left area. | The wall is far away, and the agent is moving in the lower left. |
| 3 | The agent approaches the wall from the right, but it is still some distance away. | The wall is far away, and the agent is moving in the lower right. |
| 4 | The agent is going through the gap in the middle and left, and insert the key at the end of the scene. | The agent going through the final gap and inserting the key. |
| 5 | The agent approaches the wall from the middle area. | We can identify the outline of the agent in the lower middle. |
| 6 | The agent going through the gap on the right, and performs policy far from the wall in the right area. | The agent is crossing the gap in the bottom right. |
| 7 | The agent approaches the wall from the left, but it is still some distance away. | The agent is moving in the lower left, and the outline of walls is in the middle of the screen. |

positioning to walls and gaps. This suggests that the agent has learned critical factors within the environment.

## E    HOVERING EXAMPLES

In figures E.1, and E.2, we present examples of our interactive visualization tool applied to Jumper and FruitBot. This tool is included in the supplementary material, allowing readers to freely explore the semantic distribution of features and gain a better understanding of the semantic clustering properties of DRL.

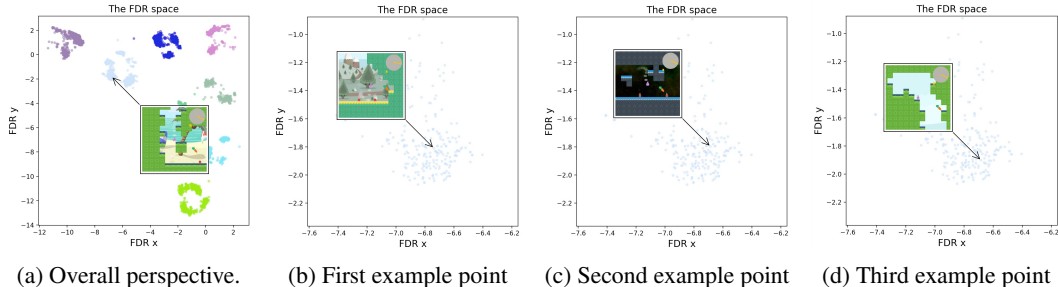

(a) Overall perspective.    (b) First example point    (c) Second example point    (d) Third example point

Figure E.1: Hover examples in the FDR space of Jumper. We observe a sub-cluster in the FDR space as an example from the overall perspective (a) and the zoomed-in perspective (b), (c), and (d). The agent is standing on the edge of a ledge. Although the scenarios of (b), (c), and (d) are different, the proposed method effectively clusters semantically consistent features together in the FDR space.

## F    SEMANTIC FORMATION IN CLUSTERS

To provide insight into the generation of semantic clustering in the feature space, we conducted the statistical analysis as follows. Initially, we collected 50,000 states using the trained model and computed the mean image for each cluster. We then calculated the mean and standard deviation of the pixel distances between the states in each cluster and their corresponding mean image. These

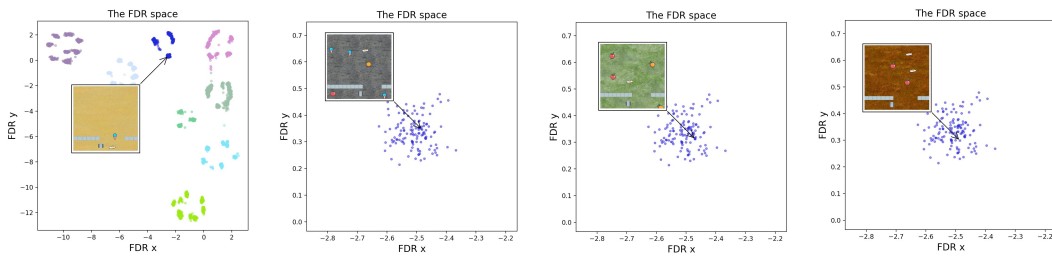

(a) Overall perspective. (b) First example point (c) Second example point (d) Third example point

Figure E.2: Hover examples in the FDR space of Fruitbot. We observe a sub-cluster in the FDR space as an example from the overall perspective (a) and the zoomed-in perspective (b), (c), and (d). The agent is standing on the edge of a ledge. Although the scenarios of (b), (c), and (d) are different, the proposed method effectively clusters semantically consistent features together in the FDR space.

Table 5: Comparison metrics of the semantically trained model verse the raw model on various environments.

| Model | Cluster transition probability | Pixel distance mean (Std. Dev.) |
|---|---|---|
| FruitBot Trained | 0.1081 | 100.00 (71.33) |
| FruitBot Raw | 0.2834 | 77.10 (49.94) |
| Jumper Trained | 0.2224 | 110.29 (62.29) |
| Jumper Raw | 0.5829 | 104.57 (58.31) |
| Ninja Trained | 0.2680 | 141.50 (67.88) |
| Ninja Raw | 0.2712 | 87.61 (62.43) |

measurements were averaged across the eight clusters. Additionally, we evaluated the probability of cluster transitions in these 50,000 states. We repeated the same statistical analysis on the 50,000 states using the untrained model, and the results are presented in Table 5. In the trained models, the pixel differences between the states within clusters increased, indicating the presence of states with similar semantics but larger pixel distances. Furthermore, the probability of cluster transitions decreased as it approached the required transition in the scene.

## G HUMAN EVALUATION DETAILS

### G.1 PART 1: OVERVIEW AND TIMELINE INTRODUCTION

This section provides a detailed description of the evaluation process and presents the timeline for conducting the assessment. The evaluation aims to assess the clarity and interpretability of the semantic clusters in the FruitBot, Jumper, and Ninja games, with the objective of enhancing and quantifying the explainability of the DRL system.

#### G.1.1 TIMELINE

Each participant will complete a survey for two game environments (FruitBot, Jumper, or Ninja). They follow the format as detailed:

- **Stage 1: Questionnaire (5 minutes)**
  Participants are requested to complete a questionnaire that collects demographic information and gaming-related details. The questionnaire includes sections for gender, age group, education level, occupation, gaming experience, familiarity with evaluating game states, and preferred game genres.

- **Stage 2: Introduction to Game Environment (10 minutes)**
  During this stage, participants receive an introduction to the evaluation process. They are informed about the objectives of the assessment and the significance of evaluating the clarity

and interpretability of the semantic clusters. They are also given a short description of the game environment (FruitBot, Jumper, or Ninja), and are shown a short gameplay clip to aid with the understanding of the game's objectives and features.

- **Stage 3: Assessment (50 minutes)**
  Following the familiarization period, participants spend 50 minutes assessing the semantic clusters in the game environment. They focus on evaluating the clarity and understandability of the video clips within each semantic cluster. This is done online via a survey.

Total Evaluation Time: 60 minutes.

## G.2 PART 2: QUESTIONNAIRE

This section of the evaluation plan presents the questionnaire that participants are required to complete. The questionnaire consists of the following sections:

### G.2.1 DEMOGRAPHIC INFORMATION

- **Age**: Participants indicate their age.
- **Gender**: Participants specify their gender as Male, Female, or Other.
- **Education Level**: Participants indicate their highest level of education completed, including options such as High school and below, Bachelor's degree, Master's degree, and Doctorate and above.
- **Occupation**: Participants provide their current occupation, selecting from options such as Student, Employee, Self-employed, or Other.

### G.2.2 GAMING-RELATED INFORMATION

- **Gaming Experience**: Participants indicate their level of gaming experience, choosing from options such as Beginner, Intermediate player, Advanced player, or Professional player.
- **Game Frequency**: Participants indicate their frequency of gaming, choosing from options such as Daily, Several times a week, Weekly, Monthly, or others.
- **Experience in Evaluating Game States**: Participants assess their experience in evaluating game states, selecting from options such as No experience, Some experience, Moderate experience, or Extensive experience.
- **Preferred Game Genres**: Participants specify their preferred game genres, including options such as Role-playing games, Shooting games, Strategy games, Puzzle games, or Other.

## G.3 PART 3: EVALUATION QUESTIONS FOR CLARITY ASSESSMENT

In this section, a comprehensive set of questions is provided to assess the clarity and understandability of the semantic clusters. The questions capture participants' opinions and perceptions using a Likert scale ranging from 'Strongly Disagree' to 'Strongly Agree'. The specific evaluation questions for the clarity assessment include:

- The clips of each cluster consistently display the same skill being performed.
- The clips of each cluster match the given skill description.

The two questions are asked each time the participant has been shown a semantic cluster.

## G.4 PART 4: EVALUATION QUESTIONS FOR INTERPRETABILITY ASSESSMENT

This section outlines the question evaluating the interpretability of the semantic clusters in terms of their usefulness. The question is designed to capture participants' opinions and perceptions using a Likert scale ranging from "Strongly Disagree" to "Strongly Agree." The specific evaluation question for the interpretability assessment:

- The identified skills aid in understanding the environment and the AI's decision-making process.

The above question is asked after the participants have seen all the semantic clusters.

## G.5 PART 5: PERSONNEL AND COORDINATION

This section outlines the personnel and coordination aspects of the evaluation. It includes information about evaluator recruitment and compensation. Specifically:

### G.5.1 EVALUATOR RECRUITMENT

15 evaluators are recruited to participate in the evaluation.

### G.5.2 EVALUATOR COMPENSATION

Each evaluator receives $15 in compensation for their valuable time and contribution to the evaluation process.

By implementing this comprehensive evaluation plan, we gather valuable insights into the clarity and interpretability of the semantic clusters in the FruitBot, Jumper, and Ninja games. The evaluation results provide essential guidance for further quantifying the improved interpretability of DRL models using our proposed method.

## G.6 PART 6: GROUPING DETAILS

- **Evaluator 1**: FruitBot, Jumper
- **Evaluator 2**: FruitBot, Jumper
- **Evaluator 3**: FruitBot, Jumper
- **Evaluator 4**: FruitBot, Jumper
- **Evaluator 5**: FruitBot, Jumper
- **Evaluator 6**: Jumper, Ninja
- **Evaluator 7**: Jumper, Ninja
- **Evaluator 8**: Jumper, Ninja
- **Evaluator 9**: Jumper, Ninja
- **Evaluator 10**: Jumper, Ninja
- **Evaluator 11**: Ninja, FruitBot
- **Evaluator 12**: Ninja, FruitBot
- **Evaluator 13**: Ninja, FruitBot
- **Evaluator 14**: Ninja, FruitBot
- **Evaluator 15**: Ninja, FruitBot

This grouping plan ensures that each evaluator evaluates two different games, and each game receives a total of 10 evaluations. It allows for comprehensive evaluations of each game and ensures that evaluators have an opportunity to provide feedback on multiple games.

# H POTENTIAL NEGATIVE SOCIETAL IMPACTS

As far as the authors are aware, there are currently no known potential negative societal impacts associated with this work.

# I  IMPACT OF THE NUMBER OF VQ EMBEDDINGS ON PERFORMANCE AND INTERPRETABILITY

To analyze the effect of the number of VQ embeddings ($K$) on both model performance and interpretability, we conducted experiments using the *Jumper* environment as an example. Similar conclusions can be extended to other environments.

## I.1  PERFORMANCE ANALYSIS

Figure I.1 shows the performance of our model with varying numbers of VQ embeddings. The results demonstrate that the number of embeddings does not affect model performance. This is expected, as our proposed method primarily focuses on feature dimensionality reduction and clustering. Combined with the overall performance results in Figure B.1, we observe that the model maintains consistent performance.

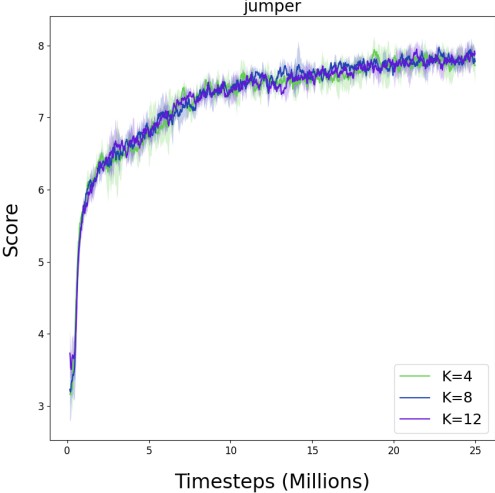

Figure I.1: Performance comparison of models with different numbers of VQ embeddings in the Jumper environment.

## I.2  INTERPRETABILITY ANALYSIS

Figures I.2a and I.2b illustrate the FDR results for $K = 4$ and $K = 12$, respectively. Our method effectively produces clusters that are clearly separable, regardless of the VQ embedding number.

However, interpretability is influenced by the choice of $K$. When $K = 12$, the semantic clusters become overly fragmented, making it difficult to form coherent semantic explanations for clusters. Conversely, when $K = 4$, Table 6 shows that clusters contain multiple distinct semantic explanations, which negatively impacts interpretability.

Clusters with incomplete or incoherent semantic descriptions hinder interpretability by introducing ambiguity in understanding the agent's behavior. This lack of clarity complicates policy analysis and makes it challenging to draw meaningful insights. Conversely, when a single cluster contains multiple interpretable behaviors, it increases the cognitive load for users who must disambiguate between these behaviors. Such a many-to-one mapping between behaviors and clusters undermines the straightforward identification of the agent's current strategy, reducing the utility of clustering as a tool for decision-making. To address these challenges, it is essential to ensure a one-to-one mapping between clusters and explanations. When each cluster is associated with a single, coherent explanation, it eliminates the need for further distinctions within clusters, facilitating clear policy analysis and enhancing human understanding of the agent's behavior.

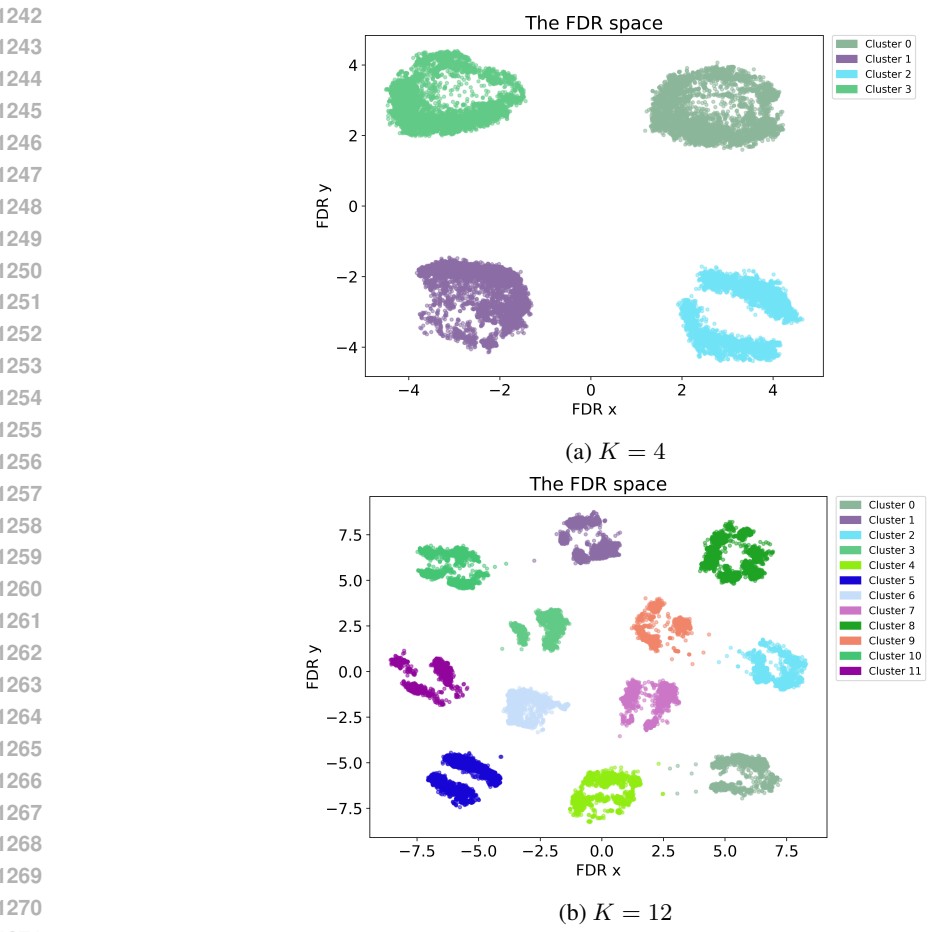

(a) $K = 4$

(b) $K = 12$

Figure I.2: Visualization of the FDR spaces for different numbers of VQ embeddings in the Jumper environment.

Table 6: Cluster descriptions for the Jumper game with $K = 4$

| Cluster | Description |
|---------|-------------|
| 0 | 1) The agent is touching the carrot on the upper left. 2) The agent is touching the carrot on the right. 3) The agent is touching the carrot on the bottom right. 4) The agent is moving in the left or lower-left part of the scene. |
| 1 | 1) The agent is touching the carrot above. 2) The agent is touching the carrot on the left. 3) The agent is moving in the right or lower-right part of the scene. |
| 2 | 1) The agent is touching the carrot below. 2) The agent is moving in the upper part of the scene. |
| 3 | 1) The agent is moving at the bottom of the scene. 2) The agent is approaching the carrot above. |

