# OpenReview forum: "Enhancing Interpretability in Deep Reinforcement Learning through Semantic Clustering"
_ICLR.cc/2025/Conference — Submitted to ICLR 2025_

### Official Review · Reviewer_WXeP · 2024-11-03

**Soundness:** 4
**Presentation:** 4
**Contribution:** 4
**Rating:** 6
**Confidence:** 3

**Summary:**

This paper investigates the semantic clustering properties of deep reinforcement learning (DRL) to enhance interpretability and understanding of internal semantic organization. The authors propose a novel DRL architecture with a semantic clustering module, integrating feature dimensionality reduction and online clustering. This module addresses the instability of t-SNE and reduces the need for manual annotation in previous methods. Experiments validate the module's effectiveness in revealing semantic clustering within DRL and introduce new analytical methods. These methods provide insights into the hierarchical structure of policies and feature space semantic organization, helping identify potential risks and guide future improvements.

**Strengths:**

1. Nice idea. The paper introduces a novel semantic clustering module within a deep reinforcement learning (DRL) architecture, which is a significant advancement in the field.

2.  The paper is well-structured and written in a clear, concise manner, making it easy to understand the core contributions and findings of the paper.

3.  The paper's experimental validation is robust and performs well.

**Weaknesses:**

1. The explanation of the clustering process lacks clarity. Specifically, it is unclear how the number of clusters, K, is determined in the context of the semantic clustering module and vq-vae. Additionally, the paper does not provide sufficient details on whether experiments were conducted to validate the choice of K or how different values of K might impact the performance and interpretability of the model. A more detailed discussion and experimental validation regarding the determination and impact of K would strengthen the overall contribution of the paper.
2. Section  4 demonstrates  that  the  paper's  experimental  validation  is  robust  and  yields  promising  results.  However,  the  paper  lacks  a comprehensive  comparison  with  existing  methods in  semantic  clustering  and  DRL interpretability.

**Questions:**

See weakness.

---

> ### Author Response · Authors · 2024-11-21
> **Response to Reviewer WXeP's Feedback**
>
> # Response to Reviewer WXeP's Feedback
>
> We sincerely thank you for the thoughtful feedback and for recognizing the significance of our semantic clustering module as a novel advancement in DRL, as well as appreciating the clear structure, concise writing, and robust experimental validation of our work.
>
> We welcome further feedback and are happy to provide responses and make improvements to the paper.
>
> ---
>
> ## Paper Revision and Line Number Changes
> We have revised the paper, resulting in changes to the line numbers. In our responses, we will use **"original"** and **"revised"** to differentiate the line numbers in the two versions.
>
> ---
>
> ## Q1: Clarity of the Clustering Process and Selection of $K$
>
> **Question**: The explanation of the clustering process lacks clarity. Specifically, it is unclear how the number of clusters, \(K\), is determined in the context of the semantic clustering module and VQ-VAE. Additionally, the paper does not provide sufficient details on whether experiments were conducted to validate the choice of \(K\) or how different values of \(K\) might impact the performance and interpretability of the model. A more detailed discussion and experimental validation regarding the determination and impact of \(K\) would strengthen the overall contribution of the paper.
>
> **Answer**:
> 1. **Clustering Process and Role of VQ-VAE**:
>    - Revised Section 3.2 (Modified VQ-VAE Loss) clarifies that VQ embeddings are used for clustering. (*revised: lines 205-207*)
>
> 2. **Selection of $K$**:
>    - The second point of Section 5 detailed our methodology for determining the number of clusters $K$. (*original: lines 468-472; revised: lines 516-520*)
>
> 3. **Impact of $K$ on Model Performance and Interpretability**:
>    - Similar to other interpretability-focused studies in the "Interpretability of DRL" section, our research prioritizes understanding the model’s decision-making process over performance optimization or parameter analysis.
>    - We validated that our model maintains performance close to the baseline, as shown in Figure B.1, ensuring that clustering does not degrade effectiveness.
>    - Additionally, we included Appendix I analyzing the impact of $K$ on both performance and interpretability. While $K$ does not affect performance, it influences interpretability.
>
> ---
>
> ## Q2: Comparison with Existing Methods in Semantic Clustering and DRL Interpretability
>
> **Question**: Section 4 demonstrates that the paper's experimental validation is robust and yields promising results. However, the paper lacks a comprehensive comparison with existing methods in semantic clustering and DRL interpretability.
>
> **Answer**:
>
> 1. **How Existing Work did**:
>    - As discussed in Lines 52–53, prior related studies relied on t-SNE visualizations to manually analyze semantic distributions in the feature space.
>
> 2. **How Clustering is Evaluated**:
>    - In Section 4.1, we evaluate the effectiveness of our clustering approach under varying input conditions, compared to t-SNE. Unlike t-SNE-based methods, our approach ensures automated and stable clustering across the feature space.
>
> 3. **How Interpretability is Evaluated**:
>    - **Interpretability Studies**: Interpretability studies often rely on case studies or human assessments rather than comparisons (e.g., works cited in "Interpretability of DRL" in Section 2) to enhance human understanding by explaining a model's decision-making process or unveiling its internal structures.
>    - **Our Work**: Our work improves interpretability by revealing semantic clustering properties and introducing novel methods for policy and model analysis.
>    - **Qualitative Analysis**: Mean images in Figures 3, D.2, D4, and explanations in Tables 1, 3, 4 present semantic groupings of states and their associated policy behaviors, using mean images and textual explanations to illustrate interpretability.
>    - **Quantitative Analysis**: Table 2 presents human evaluation results, offering quantifiable evidence that our method enhances model interpretability by aligning with human understanding.
>
> 4. **Why Direct Comparison with t-SNE-Based Interpretability is Not Feasible**:
>    - **Instability of t-SNE**: The inherent instability of t-SNE prevents it from producing reproducible clustering outcomes, making it challenging to form a solid basis for interpretability analysis (Section 4.1).
>    - **Lack of Clear Clusters**: t-SNE often fails to form distinct and separable clusters, as shown in Figure 2a. This necessitates manual annotation, as seen in related studies, which introduces subjectivity and makes fair comparisons difficult.
>    - **Dispersion of Semantic Clusters**: Due to its crowding avoidance property, t-SNE disperses cohesive semantic clusters, leaving some semantic clusters fragmented or incomplete, as shown in Figure 2b, 2c, 2f. This limitation explains why previous works annotated only portions of the feature space, as they could not derive complete semantic meanings for all clusters.

---

### Official Review · Reviewer_2Qxx · 2024-11-03

**Soundness:** 1
**Presentation:** 1
**Contribution:** 2
**Rating:** 5
**Confidence:** 3

**Summary:**

This paper proposes a novel semantic clustering method for deep reinforcement learning to improve its interpretability.

**Strengths:**

This paper proposes a semantic clustering module to cluster the state features, which uses vector quantizer to cluster the state features into some clusters.

Experiments provide cluster effectiveness evaluations as well as model and policy analysis.

**Weaknesses:**

1. The details of the algorithm are not provided. For example, how to integrate the VQ codes with the state features? Will the VQ embedding e_k be used for DRL. The paper needs more mathematical formulas to explain the process how the algorithm performs.

2. The importance of semantic clustering for DRL is not explained. Why clustering the state features can benefit DRL?

3. The enhancement of the interpretability needs to be evaluated and shown in the experiments. Could the method help the system to provide an image or sentence as the explanations of the strategy selection for the users.

4. Are there some baselines used in the experiments? I do not see the baseline section in the manuscript.

**Questions:**

See the weaknesses above.

---

> ### Author Response · Authors · 2024-11-21
> **Response to Reviewer 2Qxx's Feedback**
>
> # Response to Reviewer 2Qxx's Feedback
>
> We sincerely thank you for the thoughtful feedback and for recognizing the strengths of our work, including the proposed semantic clustering module leveraging a vector quantizer for effective state feature clustering and the comprehensive experiments evaluating cluster effectiveness and policy analysis.
>
> We welcome further feedback and are happy to provide responses and make improvements to the paper.
>
> ---
>
> ## Paper Revision and Line Number Changes
> We have revised the paper, resulting in changes to line numbers. In our responses, we will use **"original"** and **"revised"** to differentiate the line numbers in the two versions.
>
> ---
>
> ## Q1: Algorithm Details and VQ Code Integration
>
> **Question**: The details of the algorithm are not provided. For example, how to integrate the VQ codes with the state features? Will the VQ embedding \(e_k\) be used for DRL? The paper needs more mathematical formulas to explain the process.
>
> **Answer**:
> 1. **Clarification in Figure 1**:
>    - We updated the last sentence of Figure 1's caption to describe explicitly how the expanded VQ code is integrated into the state feature using addition. (*revised: lines 125-128*)
>
> 2. **VQ Embedding Usage**:
>    - Revised Section 3.2 (Modified VQ-VAE Loss) clarifies that VQ embeddings are used for clustering, which are not serving policy training. (*revised: lines 205-207*)
>
> 3. **Algorithm Pseudocode**:
>    - Section 3.2 now includes pseudocode detailing the training process, including how VQ codes integrate with state features. (*revised: lines 231-263*)
>
> 4. **Code Availability**:
>    - We will release the complete implementation to enhance reproducibility and clarity.
>
> ---
>
> ## Q2: Importance of Semantic Clustering for DRL
>
> **Question**: The importance of semantic clustering for DRL is not explained. Why clustering the state features can benefit DRL?
>
> **Answer**:
> 1. **Understanding DRL's Semantic Organization**:
>    - Semantic clustering reveals how DRL organizes states semantically, enhancing model's interpretability. (*original & revised: lines 17-19, 38-42, 63-64*).
>
> 2. **Validation of Semantic Clustering Properties**:
>    - Experiments show that semantically similar states are closer in the feature space, as supported by Figures 3, 5, D.2, D.4, E1, and E2.
>
> 3. **Practical Benefits**:
>    - Semantic clustering aids in understanding the hierarchical structure of policies and diagnosing policy behaviors. (*original & revised: lines 71-74*, Figures 4 and 6, Section 4.3, and Appendix D).
>
> ---
>
> ## Q3: Evaluating Interpretability and Strategy Explanation
>
> **Question**: The enhancement of interpretability needs to be evaluated. Could the method provide an image or sentence explaining strategy selection?
>
> **Answer**:
> 1. **Challenges in Comparison**:
>    - Interpretability research aims to enhance understanding by explaining a model's decision-making process or revealing its internal structures.
>    - A unified evaluation standard is still lacking, with most studies relying on case studies or human assessments.
>    - Methods in interpretability research vary widely, such as using attention mechanisms or decomposing policies into decision trees (as discussed in "Interpretability of DRL"), making direct comparisons challenging.
>
> 2. **Evaluation of Interpretability**:
>    - **Qualitative Analysis**: Mean images in Figures 3, D.2, D4, and explanations in Tables 1, 3, 4 show semantic state groupings and policy behavior correspondences.
>    - **Quantitative Analysis**: Table 2 presents human evaluation results, offering quantifiable evidence that our method enhances model interpretability by aligning with human understanding.
>
>     We reveal DRL's semantic clustering properties and introduce new methods to explain its decision-making, enhancing interpretability.
>
> 3. **Strategy Selection Explanation**:
>    - Our method uses VQ codes to identify the semantic cluster of the current state, linking clusters to behavior descriptions (e.g., Table 1). As demonstrated in Figures 6d and 6e, cluster changes can reflect strategy shifts, transitioning from "jumping onto a lower ledge" to "jumping onto a safer, higher ledge" (Section 4.3).
>
> ---
>
> ## Q4: Use of Baselines in Experiments
>
> **Question**: Are there some baselines used in the experiments? I do not see the baseline section in the manuscript.
>
> **Answer**:
> 1. **Evaluation of Interpretability**:
>    - Our qualitative and quantitative methods for interpretability evaluation are detailed in the previous response.
>
> 2. **Performance Analysis**:
>    - Appendix B confirms our method matches Procgen baseline performance (Figure B.1), ensuring no loss from interpretability improvements.
>    - We added Appendix I further analyzes the impact of VQ embedding number on performance and interpretability.
>
> 3. **Comparison with t-SNE**:
>    - In Section 4.1, we compare our method to t-SNE, previously used for semantic analysis, demonstrating the stability and effectiveness of our approach.

---

### Official Review · Reviewer_o27x · 2024-11-04

**Soundness:** 3
**Presentation:** 2
**Contribution:** 3
**Rating:** 6
**Confidence:** 3

**Summary:**

- This paper aims to better understand the decision making process of DRL via semantic clustering.
- Authors identify the limitations of traditional clustering techniques (mainly t-SNE) and propose a novel semantic clustering module, which incorporates a Feature Dimensionality Reduction (FDR) network and a Vector Quantizer (akin to the module used in VQ-VAE or VQ-GAN) to facilitate online clustering.
- Experiments and evaluations are carried out on the Procgen benchmark

**Strengths:**

- The problem this paper aims to tackle is very clearly defined in L53~58.
- The added FDRNet seems relatively simple (not over complicated) and the usage of Vector Quantizer for the purpose of online clustering feels reasonable.
- Visualizations clearly show the benefits of the proposed clustering method. Specifically, Figure 2 makes the comparison between the proposed FDR + VQ method vs t-SNE. Also, Figure 3 and Table 1 clearly show the different semantics of each cluster.

**Weaknesses:**

- In the "Background" paragraph of Section 3, the explanation of Vector Quantized VAEs feels very out of place. A more subtle introduction to VQ is needed. More specifically, at this point in the paper, it is not clear why the audience needs to know about VQ-VAEs. There is no context for it in previous paragraphs, so it seems very out-of-the-blue.
- In L147/148, it is said that "directly clustering such high-dimensional features is challenging, ...". It is a statement that needs to be grounded, even though most people will agree that this is true. Perhaps a quick explanation with a known source would help.
- Following on the previous point, in L150/151, "instability of t-SNE" should be elaborated on. What are the instabilities of t-SNE, and what causes these issues? It would be a good idea to add a paragraph going into the details of the difficulties of high-dimensional clustering, as well as the limitations of t-SNE.
- Under Section 3.2 (FDR loss paragraph), the claim "it more effectively captures the nonlinear structure of data and preserves local neighborhood relationships compared to methods like PCA and UMAP" needs to be well grounded. Currently, this is a claim made by the authors without a source or other logical reasoning.
- More explanation is needed on the overall pipeline (Figure 1). Especially, on the addition of VQ code (k) with the state feature. The only description is that "k is integrated into the state feature ..." but this is not detailed enough. More details on why this is added back to the state feature, what effects it has, and exactly how the "expansion into a vector" happens should be explained in the text.

**Questions:**

There are a few questions in the section above.

---

> ### Author Response · Authors · 2024-11-21
> **Response to Reviewer o27x's Feedback**
>
> # Response to Reviewer o27x's Feedback
>
> We sincerely appreciate your thoughtful feedback and your recognition of the clear problem definition, the simplicity and effectiveness of our FDRNet and Vector Quantizer for online clustering, and the impactful visualizations that highlight the benefits of our proposed clustering method.
>
> We welcome further feedback and are happy to address any additional questions or suggestions to enhance the paper.
>
> ---
>
> ## Paper Revision and Line Number Changes
> We revised the paper, which changed the line numbers. In our responses, we use **"original"** and **"revised"** to differentiate line numbers in the two versions.
>
> ---
>
> ## Q1: Explanation of Vector Quantized VAEs (VQ-VAEs)
>
> **Question**: In the "Background" paragraph of Section 3, the explanation of Vector Quantized VAEs feels very out of place. A more subtle introduction to VQ is needed. More specifically, at this point in the paper, it is not clear why the audience needs to know about VQ-VAEs. There is no context for it in previous paragraphs, so it seems very out-of-the-blue.
>
> **Answer**:
>    - We added a paragraph in the "Background" to explain why VQ-VAEs are essential to our framework and connected their use to clustering tasks in our model. (*revised: lines 141-145*)
>
> ---
>
> ## Q2: Challenges of High-Dimensional Feature Clustering
>
> **Question**: In L147/148, it is said that "directly clustering such high-dimensional features is challenging, ...". It is a statement that needs to be grounded, even though most people will agree that this is true. Perhaps a quick explanation with a known source would help.
>
> **Answer**:
>    - In Section 3.1, under "Dimensionality Reduction," we included a brief explanation of challenges in clustering high-dimensional features and added references to support the claim. (*revised: lines 153-155*)
>
> ---
>
> ## Q3: Instabilities of t-SNE
>
> **Question**: Following on the previous point, in L150/151, "instability of t-SNE" should be elaborated on. What are the instabilities of t-SNE, and what causes these issues? It would be a good idea to add a paragraph going into the details of the difficulties of high-dimensional clustering, as well as the limitations of t-SNE.
>
> **Answer**:
>    - In Section 3.1, under "Dimensionality Reduction," we added an explanation of t-SNE’s instabilities and provided references to enhance understanding. (*revised: lines 159-161*)
>
> ---
>
> ## Q4: Claim about Capturing Nonlinear Structure in Section 3.2
>
> **Question**: Under Section 3.2 (FDR loss paragraph), the claim "it more effectively captures the nonlinear structure of data and preserves local neighborhood relationships compared to methods like PCA and UMAP" needs to be well grounded. Currently, this is a claim made by the authors without a source or other logical reasoning.
>
> **Answer**:
>    - We cited Abdi & Williams (2010) to explain PCA’s focus on linear data and added Lee et al. (2004) to highlight PCA’s limitations for nonlinear data. McInnes et al. (2018) shows that Student’s t-distribution outperforms UMAP in preserving local neighborhood relationships. (*revised: lines 182-184*)
>
> ---
>
> ## Q5: Explanation of VQ Code Integration in the Pipeline
>
> **Question**: More explanation is needed on the overall pipeline (Figure 1). Especially, on the addition of VQ code (k) with the state feature. The only description is that "k is integrated into the state feature ..." but this is not detailed enough. More details on why this is added back to the state feature, what effects it has, and exactly how the "expansion into a vector" happens should be explained in the text.
>
> **Answer**:
> 1. **Enhanced Explanation in Figure 1**:
>    - We updated the caption of Figure 1 to describe explicitly how the expanded VQ code $k$ is integrated into the state feature via element-wise addition. The caption also explains that this integration supports downstream tasks. (*revised: lines 125-128*)
>
> 2. **Algorithm Pseudocode**:
>    - Detailed pseudocode was added in Section 3.2 to clarify how $k$ integrates into the state feature. (*revised: lines 231-263*)
>
> 3. **Code Release**:
>    - We will release the code to ensure transparency and provide full implementation details for readers.
>
> 4. **Contextual Explanation**:
>    - In *lines 59–60*, we discussed how previous methods struggled with downstream task integration.
>    - Under "Advantages of Online Training" in Section 3.2, we further explained that integrating $k$ into the state feature enables conditional policy training, enhancing downstream task integration. (*original: 222-224; revised: lines 267-269*)
>
>     These updates provide a comprehensive explanation of $k$’s integration, purpose, and implementation, ensuring clarity and completeness.

---

### Official Review · Reviewer_8Eq8 · 2024-11-04

**Soundness:** 2
**Presentation:** 3
**Contribution:** 2
**Rating:** 5
**Confidence:** 3

**Summary:**

This paper investigates the semantic clustering properties within deep reinforcement learning to enhance interpretability and understanding of the model's internal semantic organization. Semantic clustering here refers to grouping inputs based on semantic similarity in internal representation. The authors propose a DRL architecture with a semantic clustering module that combines feature dimensionality reduction and online clustering, integrated directly into the training pipeline. Experiments confirm the module’s effectiveness, showing it reveals the semantic structure in DRL and offers analytical tools to understand policy hierarchies and feature space organization, along with identifying model limitations for further improvement.

**Strengths:**

1.	The paper introduces a new semantic clustering module that provides valuable insights into the internal structure of DRL, improving interpretability by revealing how inputs are organized semantically within the model.
2.	Extensive experiments validate the module’s effectiveness, demonstrating its capability to uncover meaningful semantic clustering within DRL and its potential as a tool for guiding model improvements.

**Weaknesses:**

1.	The method relies on Euclidean distance to measure similarity for clustering. However, in RL scenarios, minor state variations can lead to significant changes in optimal decisions. How does the proposed method address this issue to ensure meaningful clustering?
2.	In Line 221, what specifically does "online training" refer to? Could you clarify whether this pertains to training the clustering module concurrently with the DRL model or another process?
3.	Since the method aims to interpret DRL by clustering state inputs, why not directly use the DRL output as the clustering input? Intuitively, if different states yield similar DRL outputs, they might be more likely to be semantically similar.

**Questions:**

Please kindly refer to the above weaknesses.

---

> ### Author Response · Authors · 2024-11-21
> **Response to Reviewer 8Eq8's Feedback**
>
> # Response to Reviewer 8Eq8's Feedback
>
> We sincerely thank you for your thoughtful and constructive feedback. We deeply appreciate your recognition of our semantic clustering module as a powerful tool for uncovering the internal structure of DRL and improving interpretability.
>
> We welcome further feedback and are happy to address any additional questions or suggestions to enhance the paper.
>
> ---
>
> ## Paper Revision and Line Number Changes
> We have revised the paper, resulting in changes to the line numbers. In our responses, we will use **"original"** and **"revised"** to differentiate the line numbers in the two versions.
>
> ---
>
> ## Q1: Minor State Variations and Semantic Clustering
>
> **Question**: The method relies on Euclidean distance to measure similarity for clustering. However, in RL scenarios, minor state variations can lead to significant changes in optimal decisions. How does the proposed method address this issue to ensure meaningful clustering?
>
> **Answer**: Our method addresses the challenge of minor state variations in RL scenarios through the following mechanisms:
>
> 1. **Feature Dimensionality Reduction (FDR) Module**:
>    - State features are projected into a low-dimensional FDR space, where relationships are based on relative positioning rather than values. This reduces sensitivity to minor feature variations, ensuring that small changes do not significantly impact their relative distances. (*original: lines 150-154, 171-192; revised: lines 160-165, 179-201*)
>
> 2. **Vector Quantization (VQ) Clustering**:
>    - After dimensionality reduction, VQ clustering is applied, creating broad, well-separated clusters as demonstrated in Figures 2d and 2e. This robustness minimizes the impact of small feature shifts on clustering outcomes.
>
> 3. **Semantic Consistency within the Feature Space**:
>    - Semantically similar features are positioned closer together in the FDR space, as shown in Figures 3, 5, D.2, D.4, E1, and E2. Minor state variations, such as slightly different platform heights or lengths, do not cause significant semantic shifts or FDR feature deviations in clustering.
>    -  Upon acceptance, we will release the code, allowing readers to further explore these findings using the visualization tool (Figures 5, E1 and E2).
>
> 4. **Policy Robustness across Diverse Environments**:
>    - Figure 4 illustrates clustering of semantically similar states despite differences in background patterns and layouts.
>    - Mean images in Figure 3 and explanations in Table 1 further confirm the alignment of clusters with semantically consistent policy behaviors, supplemented by video evidence in the supplementary materials.
>
> ---
>
> ## Q2: Online Training Clarification
>
> **Question**: In Line 221, what specifically does "online training" refer to? Could you clarify whether this pertains to training the clustering module concurrently with the DRL model or another process?
>
> **Answer**:
> 1. **Definition of Online Training**:
>    - In the original line 221, "online training" refers to the simultaneous training of the clustering module alongside the DRL model. As the DRL model updates its policy, the clustering module processes and clusters state features in real-time. (*original: lines 151-153 and 171-172; revised: lines 161-163, 179-181*)
>
> 2. **Additional Clarifications**:
>    - To provide further detail, we have included the algorithm's pseudocode in Section 3.2 of the revised version. This addition offers a clear explanation of the training process, including how the clustering module integrates with the DRL model. (*revised: lines 231-263*)
>
> ---
>
> ## Q3: Clustering State Features vs. DRL Outputs
>
> **Question**: Since the method aims to interpret DRL by clustering state inputs, why not directly use the DRL output as the clustering input? Intuitively, if different states yield similar DRL outputs, they might be more likely to be semantically similar.
>
> **Answer**:
>
> 1. **Loss of Semantic Context**:
>    - **Action Outputs**: Similar actions can represent different semantic tasks, e.g., in FruitBot, approaching a gate versus a fruit might involve similar movements but distinct semantics.
>    - **Value Outputs**: States with different meanings can share similar values, especially in environments like Jumper, where steady progress yields same rewards.
>
> 2. **Clustering Challenges**:
>    - Action and value sequences, even when executing semantically similar behaviors, often exhibit complex dynamics, including temporal shifts, variations in sequence length, and differences in execution details. Additionally, these sequences lack effective similarity metrics for clustering.
>
> 3. **Feature-Based Clustering Advantages**:
>    - The spatial distribution of features reflects semantic relationships, positioning similar states closer together (e.g., Figure 3). This enables effective feature distance-based clustering, preserving semantic information across the feature space while avoiding the complexity of sequence-based clustering.

---

### Meta-Review · Area_Chair_GkEr · 2024-12-19

**Metareview:**

The paper introduces a semantic clustering module to improve the interpretability of Deep Reinforcement Learning (DRL). However, it has several shortcomings, including unclear methodological details, insufficient explanation of clustering parameter selection, lack of baseline comparisons in experiments, and limited quantitative evaluation of interpretability. Thus, the current version requires further refinement before it can be considered for publication.

**Additional Comments On Reviewer Discussion:**

The paper proposes semantic clustering for DRL interpretability but lacks clarity, comparisons, and robust evaluation, needing refinement.

---

### Decision · Program_Chairs · 2025-01-22

Reject